# OpenReview forum: "LLMScan: Causal Scan for LLM Misbehavior Detection"
_ICLR.cc/2025/Conference — Submitted to ICLR 2025_

### Official Review · Reviewer_yeA7 · 2024-10-28

**Soundness:** 3
**Presentation:** 3
**Contribution:** 3
**Rating:** 5
**Confidence:** 3

**Summary:**

The paper presents LLMScan, a novel approach for detecting various types of misbehavior in large language models (LLMs). LLMScan is composed of two key components: a scanner and a detector. The scanner assesses the influence of each token through attention scores and evaluates the contribution of each layer by examining differences in output logits. These token influences and layer contributions are used to form a causal map and serve as features, which are then processed by a multi-layer perceptron (the detector) to identify misbehavior in the LLM. The paper demonstrates the effectiveness of LLMScan by detecting four types of misbehavior across 13 public datasets using three well-known LLMs.

**Strengths:**

1. The observation that the difference in causal maps between truthful and untruthful responses is significantly important for advancing research on the robustness and security of LLMs.

2. The experiments are sufficient, and the results show the effectiveness of the LLMScan.

**Weaknesses:**

The paper leverages the ATE (Average Treatment Effect) to evaluate the causal effect of each token and each layer ($ATE=E[Y|do(T=1)]-E[Y|do(T=0)]$).

My concern is replacing a token with a placeholder "-" and calculating the difference in attention scores may not fully align with the rigorous causal inference. Here are some reasons:

- In the context of causal inference, when we conduct the intervention on a token, we should expect the downstream effects on the intervention, leading to the change of other tokens. That said, this paper implicitly makes a strong assumption that all tokens are independent of each other. However, this assumption may not be held in NLP.

- From the perspective of estimating causal effect, the method only measures a difference in attention scores given a single instance, which is actually more like deriving a counterfactual sample if we assume tokens are independent.

Note that I was not questioning the effectiveness of this method. I just don't think the technique described constitutes a rigorous causal analysis. The proposed approach provides useful insights, but labeling it as causal analysis without following the established principles can be misleading.

**Questions:**

It is clear that the detector (binary classifier) can effectively identify whether a prompt can lead to misbehavior in LLM. I am curious whether the detector can be extended to precisely point out which misbehavior the LLM has given a malicious prompt.

---

### Official Review · Reviewer_LvBt · 2024-10-31

**Soundness:** 1
**Presentation:** 1
**Contribution:** 2
**Rating:** 3
**Confidence:** 4

**Summary:**

The paper proposes a new method called LLMScan for detecting misbehavior in large language models (LLMs) usage. LLMScan uses causality analysis to identify the LLM's internal activation pattern to indicate potential misbehavior, such as generating untruthful, biased, harmful, or toxic responses. The authors demonstrate the effectiveness of LLMScan through experiments on various LLMs and tasks, achieving high accuracy in detecting different types of misbehavior.

**Strengths:**

- A proactive solution by identifying and preventing the intent to generate misbehavior of LLM based on internal patterns before a token is generated is a good idea.
- The idea of analyzing internal patterns is interesting.
- The idea of constructing a causality distribution map for token and layer is interesting

**Weaknesses:**

No baseline.
The work claims to provide many baselines, but they are not presented in the relevant experiment result tables.


Poor presentation.
- The terminology provided in this work is not precise, leading to confusion for the reader. E.g.,
    - The name of causality seems misleading. The process described in ``Computing the causal effects of tokens'' is a measurement of sensitivity to token replacement by the attention scores. Since the target is an internal attention score, the term causality seems not to be an appropriate description.
    - The term causal map is also weird. The causal map provides a vector value and not a 2-dimensional matrix value.
- Besides the example of {“Who developed Windows 95?”. The truthful response is “Microsoft Corporation”. The untruthful response is “Bill Gates”} (Line 263) is also very curious since Bill Gates is the CEO of Microsoft Corporation at that time.
- Other typo/errors:
     - Table 2 duplicate first and second row

**Questions:**

Can more sophisticated causal analysis be applied for LLM detection?

---

### Official Review · Reviewer_p6Zo · 2024-11-04

**Soundness:** 2
**Presentation:** 2
**Contribution:** 2
**Rating:** 3
**Confidence:** 4

**Summary:**

This paper introduces LLMSCAN, a novel method for detecting various types of misbehavior in Large Language Models (LLMs) through causal analysis of the models' internal mechanics. The approach monitors both input tokens and transformer layers to create causal maps that capture how different components contribute to the model's outputs.

**Strengths:**

Pros:
1. The paper presents a unified approach to detecting multiple types of LLM misbehavior, whereas previous work typically focused on individual issues.
2. The authors provide detailed visualizations and analysis showing how different patterns in the causal maps correspond to different types of misbehavior, as demonstrated in Figures 2-5 and the extensive experimental results in Section 4.
3. The method shows impressive performance, particularly for detecting lies, jailbreaks, and toxic content, with AUC scores consistently above 0.95 across different models and datasets. This is particularly notable given that the method works with the first generated token, allowing for early detection of potential misbehavior.

**Weaknesses:**

Cons:
1. The performance on bias detection is notably weaker than other tasks, with AUC scores ranging from 0.71 to 0.78. While the authors acknowledge this limitation and provide some analysis in Section 4.3, they could have explored more deeply why their approach struggles with this particular type of misbehavior and potential solutions.
2. While the authors provide a public code repository, the reproducibility section (Appendix C) could be more detailed, particularly regarding the specific hyperparameters used for the detector training and the process for selecting attention heads for token-level analysis.
3. The evaluated LLMs are relatively limited and small-scale. It is suggested that the authors also evaluate on models with a larger scale such as llama-70B. Otherwise, the effectiveness of the proposed method is relatively limited.

**Questions:**

See Weaknesses

---

### Meta-Review · Area_Chair_4unE · 2024-12-16

**Metareview:**

After reading the reviewers' comments and reviewing the paper, we regret to recommend rejection.

The paper introduces a scan for detecting LLM misbehaviour called LLMScan, based on causality analysis of the model’s inner workings. While the idea is interesting, there are several technical issues that have been clearly highlighted by the reviewers and not addressed by the authors (in fact, no author response was received).
Notably, the following critical issues were identified:

1.	The performance on bias detection is notably weaker than other tasks, with AUC scores ranging from 0.71 to 0.78. While the authors acknowledge this limitation and provide some analysis in Section 4.3, they could have explored more deeply why their approach struggles with this particular type of misbehavior and potential solutions.
2.	While the authors provide a public code repository, the reproducibility section (Appendix C) could be more detailed, particularly regarding the specific hyperparameters used for the detector training and the process for selecting attention heads for token-level analysis.
3.	The evaluated LLMs are relatively limited and small-scale. It is suggested that the authors also evaluate on models with a larger scale such as llama-70B. Otherwise, the effectiveness of the proposed method is relatively limited.
4.	The terminology provided in this work is not precise, leading to confusion for the reader. E.g.,
a.	The name of causality seems misleading. The process described in ``Computing the causal effects of tokens'' is a measurement of sensitivity to token replacement by the attention scores. Since the target is an internal attention score, the term causality seems not to be an appropriate description.
b.	The term causal map is also weird. The causal map provides a vector value and not a 2-dimensional matrix value.
5.	Besides the example of {“Who developed Windows 95?”. The truthful response is “Microsoft Corporation”. The untruthful response is “Bill Gates”} (Line 263) is also very curious since Bill Gates is the CEO of Microsoft Corporation at that time.
6.	Other typo/errors:
a.	Table 2 duplicate first and second row
7.	The paper leverages the ATE (Average Treatment Effect) to evaluate the causal effect of each token and each layer (). My concern is replacing a token with a placeholder "-" and calculating the difference in attention scores may not fully align with the rigorous causal inference. Here are some reasons:

a.	In the context of causal inference, when we conduct the intervention on a token, we should expect the downstream effects on the intervention, leading to the change of other tokens. That said, this paper implicitly makes a strong assumption that all tokens are independent of each other. However, this assumption may not be held in NLP.
b.	From the perspective of estimating causal effect, the method only measures a difference in attention scores given a single instance, which is actually more like deriving a counterfactual sample if we assume tokens are independent.

**Additional Comments On Reviewer Discussion:**

The reviewers raised several concerns that were not addressed by the authors (no response received).

No ethics review raised by the reviewers, and we agree with them.

---

### Decision · Program_Chairs · 2025-01-22

Reject